# Maternal Iron Deficiency and Environmental Lead (Pb) Exposure Alter the Predictive Value of Blood Pb Levels on Brain Pb Burden in the Offspring in a Dietary Mouse Model: An Important Consideration for Cumulative Risk in Development

**DOI:** 10.3390/nu15194101

**Published:** 2023-09-22

**Authors:** Janine Cubello, Derick R. Peterson, Lu Wang, Margot Mayer-Proschel

**Affiliations:** 1Department of Environmental Medicine, University of Rochester, Rochester, NY 14642, USA; janine_cubello@urmc.rochester.edu; 2Department of Biostatistics and Computational Biology, University of Rochester, Rochester, NY 14642, USA; derick_peterson@urmc.rochester.edu (D.R.P.); lu_wang@urmc.rochester.edu (L.W.); 3Department of Biomedical Genetics, University of Rochester, Rochester, NY 14642, USA

**Keywords:** micronutrients, metals, anemia, pregnancy, Swiss Webster, Pb burden, neurodevelopment, risk assessment

## Abstract

Maternal iron deficiency (ID) and environmental lead (Pb) exposure are co-occurring insults that both affect the neurodevelopment of offspring. Few studies have investigated how ID affects brain-region-specific Pb accumulations using human-relevant Pb concentrations. Furthermore, how these Pb exposures impact blood and brain Fe levels remains unclear. Importantly, we also wanted to determine whether the use of blood Pb levels as a surrogate for the brain Pb burden is affected by underlying iron status. We exposed virgin Swiss Webster female mice to one of six conditions differing by iron diet and Pb water concentration (0 ppm, 19 ppm, or 50 ppm lead acetate) and used Inductively Coupled Plasma Mass Spectrometry to measure the maternal and offspring circulating, stored, and brain Pb levels. We found that maternal ID rendered the offspring iron-deficient anemic and led to a region-specific depletion of brain Fe that was exacerbated by Pb in a dose-specific manner. The postnatal iron deficiency anemia also exacerbated cortical and hippocampal Pb accumulation. Interestingly, BPb levels only correlated with the brain Pb burden in ID pups but not in IN offspring. We conclude that ID significantly increases the brain Pb burden and that BPb levels alone are insufficient as a clinical surrogate to make extrapolations on the brain Pb burden.

## 1. Introduction

According to the World Health Organization (WHO), iron deficiency (ID) is the most prevalent micronutrient deficiency in the world, encompassing roughly half of all anemia cases, and is highly prevalent amongst pregnant women and individuals residing in areas of lower socioeconomic status [1]. Recent studies have shown that 15–20% of pregnant women experience severe iron depletions in the form of iron deficiency anemia (IDA) and an even greater percentage, 20–50%, develop a non-anemic ID during their first trimester [2,3].

The impact of maternal ID on offspring neurodevelopmental outcomes is well established (i.e., [4,5,6,7]) and likely exacerbated by various socioeconomic factors such as food insecurity tied to lower socioeconomic status [8], postnatal stress [9], and exposure to environmental Lead (Pb). Despite governmental regulations to minimize Pb exposure and an overall decline in BPb levels in the general population within the United States over the years [10,11,12], these regulations have not fully addressed the sources of Pb contamination in older homes and drinking water systems still present in these communities [13,14,15,16], creating a health disparity that raises considerable public concern [17,18,19,20,21,22,23,24,25,26]. Upon exposure, the majority of Pb is preferentially stored in mineralized tissues such as bone [27]. Importantly, increases in maternal bone resorption during pregnancy to support fetal calcium needs for bone development can mobilize maternal Pb stores into circulation, rendering the developing offspring susceptible to both past and present Pb exposures [28,29,30,31]. Once in circulation, Pb transfers to the developing offspring [32] and competes with Fe in its utilization of homeostatic machineries for mobilization and uptake [33,34,35].

Expanding upon our existing understanding of the associations between maternal ID and Pb exposure is crucial because these exposures do not occur in isolation. The potential for ID and Pb exposure to act as disease modifiers for one another, especially during neurodevelopment [36,37,38,39,40], is highlighted in epidemiological studies that show the associations between IDA and higher blood Pb levels (BPb) in children [41,42,43] and a negative dose effect relationship between BPb levels and hemoglobin levels in antenatal women [44,45]. Moreover, a recent study found that Fe levels before the onset of pregnancy can set the tone for the maternal BPb levels achieved during pregnancy, as reflected by higher first-trimester BPb levels in women whose serum ferritin concentrations were lower prior to pregnancy [46]. These data support the idea that both ID and Pb can alter the typical clinical measures used to assess the risk associated with either insult alone, and there is value in investigating these insults as a co-exposure. Furthermore, although recent reductions in the BPb reference value (5 µg/dL to 3.5 µg/dL) by the Centers for Disease Control and Prevention (CDC) [47] reflect concerns that there is no “safe” concentration of Pb exposure, the vulnerability of the developing brain to even low Pb concentrations is evident with the presence of neurodevelopmental, cognitive, and attention deficits at BPb levels below 5 µg/dL [48,49,50,51,52,53]. Aside from being clinically attainable, the usage of BPb levels in extrapolating Pb risk, particularly in the brain, is based on the underlying assumption that increased circulating blood Pb will increase the brain Pb burden and elicit direct neurotoxic actions on brain cells (for review [54]). This assumption, however, is contingent on a correlation between blood and brain Pb burdens, the validity of which. to our knowledge, has not been validated for specific brain regions or in the context of ID.

In light of these knowledge gaps, we established a novel co-exposure model merging a highly controlled dietary paradigm of ID with life-long, human-relevant concentrations of Pb. This model allowed us to address the following questions: (i) Does maternal ID and/or low-level Pb exposure affect offspring blood and brain iron status? (ii) Does maternal ID exacerbate offspring Pb burdens in blood, the femur, and the brain otherwise seen if on an iron-normal (IN) diet? (iii) Are blood Pb levels a useful surrogate for estimating the offspring brain Pb burden in the absence or presence of ID? Using Inductively Coupled Plasma Mass Spectrometry (ICP-MS), we quantified the Fe and/or Pb levels in the whole blood (circulating) and femurs (stored) of exposed dams and their offspring at postnatal days (PND) 6-9. In addition, we analyzed the PND6-9 offspring brain Fe and Pb burden in the cerebral cortex and hippocampus, which are known to undergo neurodevelopmental disruptions as a consequence of ID or Pb exposure alone [51,55,56,57,58,59,60,61] and correlated the BPb levels to the brain Pb burden to determine the predictive value of blood Pb levels for estimating the brain Pb burden, and thereby, neurodevelopmental risk.

## 2. Materials and Methods

### 2.1. Animals and Study Design of Diet and Pb Exposures

All animal exposures, handling, and tissue harvests were conducted by the same individual (JC) to ensure a consistency in the dissections and a minimization of stress to the animals. All protocols were approved prior to this study by the University Committee on Animal Resources at the University of Rochester (Protocol #: 2001-240E). Six- or ten-week-old virgin Swiss Webster dams obtained from Charles River Laboratories (Wilmington, MA, USA) were housed under controlled temperature and humidity conditions on a 12 h light/dark cycle for the duration of the experiment. All animals were allowed to acclimate to vivarium housing for two weeks prior to any experimental manipulations. Iron-normal (IN; 240 µg Fe/g; TD. 05656) or Iron-deficient (ID; 2–6 µg Fe/g; TD. 80396) animal diets, identical in composition aside from iron levels, were purchased from Envigo (formerly Harlan Teklad, Indianapolis, IN, USA). To prevent unintentional iron supplementation, animals were housed utilizing 1/8″ Bed-o’Cobs corn cob bedding (The Andersons, Inc., Maumee, OH, USA; 8B) rather than paper bedding and a plastic house as opposed to nestlets for enrichment.

Within this study, a total of 52 eight-week-old virgin female mice were randomly assigned to one of six different exposure groups: IN diet + distilled water (0-IN; *n* = 14), ID diet + distilled water (0-ID; *n* = 12), IN diet + 19 ppm Pb-distilled water (19-IN; *n* = 4), ID diet + 19 ppm Pb-distilled water (19-ID; *n* = 4), IN diet + 50 ppm Pb-distilled water (50-IN; *n* = 11), and ID diet + 50 ppm Pb-distilled water (50-ID; *n* = 7). Note: we generated more non-Pb-exposed litters for unrelated studies (not reported here) and increased the number of dams assigned to 50 Pb to offset a potentially greater mortality rate at a higher Pb concentration. Pb water was prepared fresh weekly by dissolving lead (II) acetate trihydrate (Sigma-Aldrich, St. Louis, MO, USA; 467863) in distilled water, and Pb water was continuously administered to Pb-designated animals throughout the course of the experiment. Specifically, two months prior to mating, Pb-designated females began an oral administration of either 19 or 50 ppm Pb acetate (PbAc) drinking water (equivalent to 10 ppm and 27 ppm Pb, respectively) *ad libitum* while fed on a standard 5010 vivarium rodent diet (LabDiet, St. Louis, MO, USA; 0001326). This experimental design ensured maternal Pb loading prior to pregnancy onset, as would typically occur in humans. At twelve weeks of age (four weeks into Pb exposures for Pb groups), all females within this study (Pb and non-Pb) were transitioned from the standard vivarium diet to the IN diet. After a two-week acclimation period to the IN diet, the cohorts received their final designated diet, where IN-designated animals were maintained on the IN diet and ID-designated animals were switched to the ID diet. Two weeks later, and within each treatment group, females were paired 2:1 with males similar in size and age. The male breeders assigned to the Pb-exposed females were only exposed to Pb during mating with their corresponding dams. A summary of the exposure paradigm is shown in Figure 1.

The dams were checked daily for the birth of pups, and once observed, pups were designated as postnatal day 1 (PND1). Litter sizes were recorded at birth and pup body weights were measured at postnatal day 7. Unlike for many other studies, the resulting litters were not culled in order to minimize maternal stress and to preserve the natural intra-litter heterogeneity in the distribution of Fe and Pb across both in utero and postnatal development. A subset of dams and their corresponding offspring were sacrificed at postnatal day PND6-9 in accordance with the AVMA Guidelines for the Euthanasia of Animals. Post blood collections and prior to tissue microdissections, dams were perfused with heparin/1X PBS (Sigma-Aldrich, St. Louis, MO, USA; H3149), whereas pups remained un-perfused. All analyses within the treatment groups were conducted on a minimum of one pup per sex per litter, spread across 3–7 representative litters, except for litter size, which includes collective data used across endpoints from 4 to 14 litters. Some littermates were additionally used for experimental analyses outside the scope of this study and are therefore not included. For each outcome, the number of litters and offspring assessed from each litter are outlined in detail in Appendix A.

### 2.2. Hematological Analyses

At approximately one week postpartum, maternal and pup whole blood samples were collected into K2EDTA-coated BD Microtainer Tubes (BD, Franklin Lakes, NJ, USA; 365974) and stored on ice for a maximum of two hours until hematocrit and hemoglobin concentration values were assessed by an individual blinded to the cohort identities using a HESKA-HemaTrue^®^ Veterinary Hematology Analyzer (Heska Corporation, Loveland, CO, USA; 5600). Within the analyzer, hematocrit values reflect the percentage of red blood cells in whole blood (%RBC). Hemoglobin concentrations reflect the number of grams of hemoglobin per deciliter of blood (g/dL), a value measured spectrophotometrically as RBC are lysed and intracellular hemoglobin is released within the analyzer. The trunk whole blood of pups was collected via decapitation, while maternal blood was collected utilizing a heparin-coated syringe (Sigma-Aldrich, St. Louis, MO, USA; H3149) for intravenous terminal blood draws through the inferior venae cava post ketamine overdose.

### 2.3. Tissue Harvests

#### 2.3.1. Microdissection of Cortical and Hippocampal Brain Tissue

Post careful removal of the skin and skulls of decapitated pups, whole brains were micro-dissected to isolate the cerebral cortex and hippocampus in Leibovitz’s L-15 with L-glutamine (Fisher Scientific, Waltham, MA, USA; MT10045CV). To ensure the metal measurements reflected cortical metal levels only, cerebral cortices were micro-dissected to exclude nearby white matter, specifically the corpus callosum. For each individual brain region, both hemispheres were put into pre-weighed 1.7 mL polystyrene RNase, DNase, Pyrogen-free microcentrifuge tubes on ice (Laboratory Products Sales, Inc. (LPS), Rochester, NY, USA; L211511). Tissue-containing tubes were then weighed before storing at −80 °C.

#### 2.3.2. Femur

Femurs were cut above the knee joint and in-between the femoral head and neck. The remaining muscle and connective tissues were trimmed off, and individual femurs were not flushed of their bone marrow. As was the case for brain tissue, individual femurs were collected into pre-weighed microcentrifuge tubes, weighed, and stored at −80 °C.

### 2.4. Inductively Coupled Plasma Mass Spectrometry (ICP-MS)

In collaboration with the University of Rochester Elemental Analysis Facility, multi-element detections at sensitivities in the parts per billion (ppb) and parts per trillion (ppt) ranges were conducted by an individual blinded to the cohort identities using a PerkinElmer NexION 2000C ICP Mass Spectrometer (PerkinElmer, Waltham, MA, USA; NB150045). All samples were hydrolyzed in their microcentrifuge tubes in ultrapure (for trace metal analysis) 67–70% nitric acid (VWR International, Radnor, PA, USA; 87003-658) in a heat block at 100 °C for 1 h and brought up to a final volume of 10 mL in UltraPure de-ionized water. This dilution factor was accounted for and the metal levels per sample were normalized to the wet weights of freshly harvested tissue and presented as µg/dL for blood and µg/g for femur and brain tissue.

### 2.5. Statistical Analysis

For the analyses conducted within this study (i.e., hematological assessments, Fe, and Pb), we did not exclude any animals to allow for variability in the Fe and Pb responses attributed to genetic diversity within this outbred mouse strain. Gamma generalized linear models (GLMs) with a log link were used to model each outcome as a function of Pb exposure (0, 19, 50 ppm PbAc), iron status (ID vs. IN), litter size, sex, and the interaction between Pb exposure and the ID diets administered. Since the interaction was typically significant, thus complicating interpretations, the data were further stratified by either ID or Pb exposure. Specifically, we went on to fit separate GLMs to estimate the effects of Pb exposure within each iron diet group and another set of models to estimate the effects of iron diet within each concentration of Pb exposure, all adjusted for litter size and sex. When modeling iron levels, litter size, hematocrits, hemoglobin concentrations, and pup weights, we considered the effects of the Pb and iron diet across all concentrations of Pb exposure (0, 19, 50 ppm PbAc); however, when modeling Pb levels, we only compared across the 19 and 50 ppm PbAc-exposed mice. All 95% confidence intervals (CI) and *p*-values were likelihood-based, and the 0.05 level of significance was used for all hypothesis tests. For individual 19-Pb-exposed pups, where Fe and Pb values were measured in both the blood and brain tissue, Pearson’s product moment correlation was used to quantify and test the association between log(blood Pb concentration) and log(tissue Pb concentration) for each tissue (cerebral cortex and hippocampus), stratified by ID, and the corresponding least squares lines were superimposed on the data, plotted on a log–log scale. It is important to note that while blood, femur, cerebral cortex, and hippocampus tissues were generally micro-dissected from each animal, this study could not necessarily assess every outcome measurement for every animal. This was especially true due to limitations in the volumes of trunk blood attainable from PND6-9 offspring, volumes that were insufficient for appropriately measuring both the hematocrits and whole blood metal levels from the same animal. Therefore, the majority of analyses presented within this study, aside from correlation analyses, are not reflective of intra-individual responses, but rather comparisons between treatment cohorts. All computations were carried out using R version 4. Aside from the correlation scatterplots produced using R version 4, all data visualized in the graphs were plotted using GraphPad Prism.

## 3. Results

### 3.1. Maternal Iron Deficiency Is Not Exacerbated by Environmentally Relevant Pb Exposures

Knowing that pregnant women are susceptible not only to ID but also Pb exposure, we established a clinically relevant co-exposure model using environmentally relevant Pb concentrations (19 and 50 Pb) [62,63]. In our previous work, we found that prior to the significant depletions in maternal hematocrits on gestational day 16, our ID dietary paradigm reduces embryonic brain Fe levels as early as gestational day 12 [61]. Using the exposure paradigm described in Figure 1, we confirmed that in the absence of Pb, maternal exposure to our ID diet (0-ID) resulted in a significant decrease in hematocrits (*p =* 0.011) and hemoglobin levels (*p =* 0.015) but no observed changes in blood Fe levels (*p =* 0.75) one week postpartum (Figure 2A–C, open symbols). Amongst 19-Pb-exposed dams (Figure 2A–C, orange symbols), the hematocrits of the ID dams were significantly lower than those of the IN dams (*p =* 0.020), whereas there was insufficient evidence of a significant difference in their hemoglobin concentrations (*p =* 0.058) and blood Fe levels (*p =* 0.11). In contrast, in 50-Pb-exposed dams (Figure 2A–C, blue symbols), the hematocrits (*p =* 0.66), hemoglobin concentrations (*p =* 0.85), and blood Fe levels (*p =* 0.14) were comparable between the IN and ID dams.

When we stratified by diet, the effects of Pb exposure were dependent on the Pb concentration and measured outcomes. There was a significant interaction between iron diet and Pb exposure for both the hematocrits (*p =* 0.016) and hemoglobin levels (*p =* 0.007) but not blood Fe (*p =* 0.24) (Figure 2A–C).

Amongst the IN dams, there was insufficient evidence of an effect of 19 Pb or 50 Pb on the hematocrits (19 Pb: *p =* 0.24; 50 Pb: *p =* 0.72) or blood Fe levels (19 Pb: *p =* 0.44; 50 Pb: *p =* 0.11); however, the hemoglobin concentrations were significantly reduced when the dams were exposed to 19 Pb (*p =* 0.04). Interestingly, these reductions were not observed in 50-IN dams (*p =* 0.083). Amongst the ID dams, the hematocrits (*p =* 0.12), hemoglobin concentrations (*p =* 0.16), and blood Fe levels (*p =* 0.063) were not altered by 19 Pb. In 50-ID dams, however, in the absence of significant changes in the blood Fe levels (*p =* 0.15), significant increases were observed in the hematocrits (*p =* 0.002) and hemoglobin concentrations (*p =* 0.011). 

Taken together, these data show that our dietary ID model generates a mild iron deficiency anemia (IDA) that is not exacerbated by the concentration of Pb, nor do the Pb concentrations alone induce anemia in IN dams.

### 3.2. Maternal ID Increases Circulating and Stored Maternal Pb Levels

An important aspect of our co-exposure model is the consideration that the developing fetus is at risk of both past and present Pb exposures. The enhanced bone remodeling that occurs in pregnancy releases Pb stores into circulation. In the femurs collected from dams and in the same maternal blood samples analyzed in Figure 2, we used ICP-MS to confirm that the Pb concentrations administered do, in fact, increase the maternal stored Pb burden and circulating BPb levels (Figure 3A,B, respectively). The analysis of the femurs revealed an overall significant interaction (*p =* 0.027) between diet and Pb exposure (Figure 3A). Irrespective of whether the dams were exposed to 19 or 50 Pb, the ID dams had significantly greater femoral Pb levels than the IN dams (19 Pb: *p <* 0.001; 50 Pb: *p =* 0.025). Interestingly, the differences in the femoral Pb burden between the 19 and 50 Pb exposures were dependent on diet. For instance, the femoral Pb burden in the 50-IN dams was significantly greater than in the 19-IN dams (*p <* 0.001); however, there was insufficient evidence of a significant difference between the 50-ID and 19-ID dams (*p =* 0.17).

Similar to the femurs, the BPb levels were significantly greater in the ID dams compared to the IN dams at both Pb concentrations (19 Pb: *p <* 0.001; 50 Pb: *p =* 0.004). The differences in the BPb levels between the 19 and 50 Pb doses were also dependent on diet. For instance, the BPb levels in the 50-IN dams were significantly greater than in the 19-IN dams (22.6 µg/dL vs. 11.3 µg/dL, *p =* 0.002); however, this difference was not evident in the dams exposed to an ID diet (40.2 µg/dL vs. 31.6 µg/dL, *p =* 0.17) (Figure 3B).

These data suggest that, irrespective of the concentration of Pb administered, exposure to an ID diet exacerbates the levels of maternal circulating and stored Pb. In addition, dietary iron levels affect the dose-dependent distribution of Pb into different compartments.

### 3.3. Maternal ID and Pb Exposure Affect Pup Weights, but Not Litter Size

Litter size has not been reported to be affected by Pb [64,65] or mild maternal IDA [66,67,68] in isolation, but the impact of their co-exposure on litter size is not yet clear. Therefore, we analyzed the number of living pups observed at PND7 across 4–14 independent litters for each treatment group and found there was insufficient evidence that the exposure of the dams to 50 Pb or 19 Pb in the absence or presence of ID affected the litter sizes (*p* > 0.05) (Figure 4A). In fact, in the co-exposure group (50-ID), the average number of living pups across seven independent litters (8.6 ± 3.6 pups) was nearly identical to the average number of pups/litter born to both the 50-IN (8.6 ± 2.4 pups) and 0-IN dams (8.8 ± 2.5 pups), suggesting that neither single or co-exposure to our highest Pb concentration causes postnatal lethality in the offspring or significant impairments in the reproductive capacity of the exposed mothers.

Unlike litter size, weight at birth has been associated with both gestational Pb and ID in rodent models [61,68,69,70] as well as humans [71,72]. We thus analyzed the treatment-specific effects on pup weights (Figure 4B). Irrespective of whether the dams received 0, 19, or 50 Pb, the weights of the ID offspring were consistently significantly lower than those of the IN offspring (*p ≤* 0.002). Interestingly, when stratified by diet, we found that the effects of Pb concentration differed depending on the diet context. Within the IN offspring, the pup weights were not significantly altered by a 50 Pb exposure (*p =* 0.16) but were increased if exposed to 19 Pb (*p =* 0.014). In contrast, in the ID offspring, the pup weights were not significantly altered by 19 Pb (*p =* 0.36) but were reduced if exposed to 50 Pb (*p =* 0.007).

Taken together, neither diet nor Pb exposure affected litter size but the ID pups were consistently smaller than the IN pups. In addition, the effects of Pb on weight were dependent on Pb concentration and diet.

### 3.4. Maternal ID Causes an Iron Deficiency Anemia (IDA) in Offspring That Is Not Exacerbated by Pb Exposure

While the reductions in weight observed in the ID offspring do suggest the presence of an iron deficiency, we wanted to confirm the pup iron status via hematological parameters and blood Fe levels. An analysis of the pup hematocrits showed an overall significant interaction between iron diet and Pb exposure (*p <* 0.001). Within each Pb context, the ID pups consistently had significantly lower hematocrits (0 Pb, 19 Pb, 50 Pb: *p <* 0.001) and hemoglobin levels (0 Pb, 19 Pb *p <* 0.001, and 50 Pb *p =* 0.001) than the IN pups (Figure 5A,B).

When we stratified by diet, neither Pb concentration significantly affected the hematocrits in the IN pups (*p >* 0.05, Figure 5A). The HGB levels were not affected by a 50 Pb exposure (*p =* 0.50) and only marginally by a 19 Pb exposure (*p =* 0.048) (Figure 5B). In the ID pups, there was insufficient evidence that an exposure to 19 Pb significantly affected the hematocrits or HGB levels *(p >* 0.40) (Figure 5). With a 50 Pb exposure, the hematocrits were, however, significantly increased (*p =* 0.001), while the HGB levels remained unaffected (*p =* 0.095) (Figure 5A,B). These observations suggest that an exposure to 19 Pb or 50 Pb alone does not induce IDA in IN pups. In addition, decreases in pup hematocrits and hemoglobin concentrations are mainly driven by diet rather than Pb exposure.

### 3.5. Maternal ID and Pb Exposure Disrupt Offspring Circulating and Brain Iron Levels in a Region-Specific Manner

To gain a better understanding of circulating Fe levels, we performed an ICP-MS analysis of the trunk blood collected from the postnatal offspring. As shown in Figure 6A, there was insufficient evidence of an interaction between diet and Pb exposure on blood Fe levels (*p =* 0.27). Generally, the ID offspring had significantly lower circulating blood Fe than the IN offspring, irrespective of Pb exposure (0 Pb, 19 Pb, 50 Pb: *p <* 0.001). The effect of Pb exposure alone did, however, affect the circulating blood Fe levels in the IN and ID offspring in a concentration-dependent manner. Specifically, the blood Fe levels in the IN offspring were significantly reduced by exposure to 19 Pb (*p =* 0.001) but were significantly increased by exposure to 50 Pb (*p =* 0.039). In ID pups, an exposure to 19 Pb also resulted in significant decreases in blood Fe levels (*p <* 0.001); however, there was insufficient evidence of a significant change in the 50 Pb pups (*p =* 0.53). These data indicate that circulating Fe levels can be altered by both Pb exposure and ID; however, the most profound changes in circulating Fe levels occurred in the context of iron deficiency.

Maternal ID has consistently been associated with negative neurodevelopmental outcomes (i.e., [73]) and additional studies in animals [7] and humans [74] suggest that depletions of brain Fe levels play a role in these impairments. We thus tested whether, similar to blood Fe, brain Fe levels were affected by diet and Pb exposure.

We first found an interaction between diet and Pb exposure (*p =* 0.015). The cortical Fe levels in the ID offspring were significantly lower than those of the IN offspring, irrespective of the concentration of Pb exposure (Figure 6B) (0 Pb, 19 Pb, 50 Pb: *p ≤* 0.005). When we stratified by diet, in the IN offspring, 19 Pb did not affect the cortical Fe levels (*p =* 0.57); however, a significant accumulation of cortical Fe levels occurred with exposure to 50 Pb (*p =* 0.045). In contrast, in the ID offspring, the cortical Fe levels were significantly decreased when they were exposed to 19 Pb (*p =* 0.001), but there was insufficient evidence of changes in their cortical Fe levels when they were exposed to 50 Pb (*p =* 0.21).

The hippocampal sensitivities to Fe depletion did not mirror those observed in the cerebral cortex and blood (Figure 6C). In the ID pups, the hippocampal Fe levels were not significantly affected in the absence of Pb (*p =* 0.20) or the presence of 50 Pb (*p =* 0.55). However, the 19-ID pups had significantly reduced hippocampal Fe levels compared to the 19-IN pups (*p <* 0.001). Irrespective of diet, the hippocampal Fe levels were not significantly altered by exposure to 50 Pb (*p >* 0.2) but were decreased by exposure to 19 Pb (*p ≤* 0.006). These data suggest that in early postnatal offspring, region-specific sensitivities to brain Fe depletions are modulated by Pb exposure. 

### 3.6. Elevations in Pb Burden Attributed to ID Are Not Always Reflected in BPb Levels

BPb levels are currently used as a surrogate for surveying Pb risk in humans. As Fe and Pb are known to affect each other, we investigated whether and to what extent maternal ID could affect Pb levels across various tissues in the offspring. ICP-MS was used to first measure the blood and femoral Pb levels in the exposed offspring (Figure 7).

We found a significant interaction between diet and Pb exposure (*p =* 0.015) in the blood (Figure 7A). The BPb levels were significantly greater in the ID pups than the IN pups when exposed to 19 Pb (*p <* 0.001) but not 50 Pb (*p =* 0.25). When stratified by diet, the BPb levels were significantly greater with a 50 Pb exposure than a 19 Pb exposure (*p ≤* 0.001) in the IN (43.8 ± 9.8 µg/dL vs. 19.9 ± 5.8 µg/dL) and ID (48.7 ± 14.0 µg/dL vs. 30.1 ± 12.7 µg/dL) pups.

Since Pb accumulates in bone, we also measured Pb in the femurs of these offspring (Figure 7B) and found again sufficient evidence of an interaction between diet and Pb exposure (*p =* 0.034). However, unlike BPb, irrespective of the Pb concentration, the femoral Pb levels were significantly greater in the ID pups than in the IN pups (*p <* 0.001). When stratified by diet, we further found that in both the IN and ID offspring, the femoral Pb levels were significantly greater in the 50 Pb pups than in the 19 Pb pups (*p <* 0.001).

Taken together, these data suggest that the circulating and stored Pb burden is a consequence of an interaction between an individual’s Fe status and the Pb exposure they experience during development.

### 3.7. Maternal ID Contributes to Brain Pb Accumulation in Offspring

We next determined whether maternal ID and Pb exposure affect offspring brain Pb accumulation, specifically in the hippocampus and cerebral cortex. As shown in Figure 8A,B, we saw a significant interaction between diet and Pb (*p ≤* 0.011) for both regions. Additionally, irrespective of Pb concentration, the ID pups consistently had significantly greater Pb levels than the IN pups in both the cerebral cortex and hippocampus (*p ≤* 0.016). When stratified by diet, the ID pups had significantly greater brain Pb burdens with a 50 Pb exposure than a 19 Pb exposure (*p <* 0.001). In contrast, this difference was not evident in the IN pups (*p >* 0.3). These data suggest that ID enhances the cortical and hippocampal Pb accumulation otherwise seen under IN conditions.

### 3.8. Blood Pb Levels of Offspring Correlate with Brain Pb Levels in ID but Not in Offspring

Currently, the CDC guidelines advise using BPb levels to assess an individual’s Pb risk [47]. To determine whether circulating BPb levels are informative of the stored and brain Pb burden, we assessed whether BPb levels correlate with brain (cerebral cortex and hippocampus) Pb levels. We focused on correlations within the 19 Pb offspring, as the BPb levels were most reflective of lower human-relevant exposure concentrations [75]. As shown in Figure 9, in the IN offspring, the BPb levels were not correlated with the brain Pb levels (cerebral cortex r = −0.1, *p =* 0.72; hippocampus r = −0.39, *p =* 0.13). In contrast, in the ID offspring, the BPb levels were strongly correlated with the brain Pb levels (cerebral cortex: r = 0.92, *p <* 0.001; hippocampus: r = 0.76, *p =* 0.001).

Importantly, these data suggest that without knowledge of the underlying gestational history of Fe levels or consideration of the iron status of the postnatal offspring, BPb levels alone are not an informative surrogate for estimating the neurological risk associated with a given Pb exposure.

## 4. Discussion

In our study, we combined a well-defined nutritional mouse model of ID with a life-long Pb exposure and found not only that ID is a critical risk modifier in determining the tissue Pb burden at a critical window early in development, but its influence is Pb-concentration- and tissue-specific. Importantly, we showed that the strength of the correlation between blood Pb concentrations and brain Pb burdens in early postnatal mouse offspring was very dependent on whether ID was experienced. Specifically, the cortical and hippocampal Pb levels were only strongly correlated to blood Pb levels when the offspring were exposed to an ID diet during pregnancy.

We defined 50 ppm PbAc and 19 ppm PbAc (abbreviated 50 Pb and 19 Pb, respectively) as human-relevant concentrations of Pb based on published data mainly conducted in C57BL/6J mice. Interestingly, however, the BPb achieved in our IN Swiss Webster dams and offspring was on average higher than BPb levels previously reported in iron-sufficient 100 ppm PbAc-exposed C57BL/6J mice (dams: 12.6 µg/dL, pups: 10.2–12.5 µg/dL) [65,76,77]. After confirming that differences in ~PND7 BPb levels between Swiss Webster and C57BL/6J mice were not attributed to baseline diet composition (Envigo-adjusted iron diets vs. standard vivarium rodent chow, these data suggest that the differences in BPb values between our studies and others are likely a consequence of the genetic diversity and differences in mouse strain-specific susceptibilities [78].

Despite these differences in BPb levels, none of our treatment groups were significantly altered in terms of litter size, suggesting low embryonic lethality. We did, however, observe significant reductions in the weights of exposed pups but only in offspring of the mothers that received an ID diet. These findings are consistent with observations in both rodents [61,68,69,70] and humans [71,72], where reductions in maternal iron status, especially earlier in pregnancy, have been associated with lower birth weights in offspring.

Contrary to the consistent reductions in pup weights attributed to exposure to an ID diet, the effect of Pb on pup weights was not unidirectional but rather context-dependent. For example, we found that a 50 Pb exposure reduced the weights in the ID but not the IN offspring; however, a 19 Pb exposure increased the weights in the IN but not the ID offspring. This supports the conflicting findings in humans, where some report significant negative associations between prenatal Pb exposure and birth weight [79,80,81], whereas others have found no association [82,83,84].

Irrespective of the underlying mechanisms that define weight changes by diet or Pb, the decreases in the hematocrits and hemoglobin concentrations in the offspring suggested that a mild IDA in the dams resulted in the generation of a more severe iron deficiency anemia (IDA) in pups that was not further exacerbated by Pb. This could suggest the prioritization of a limited amount of available dietary Fe towards the maintenance of maternal Fe stores rather than the offspring. These observations are consistent with the findings in humans. For example, Basu et al. [85] also found that cord blood iron indices and neonatal brain development were strongly correlated to maternal iron indices, irrespective of the severity of IDA. In addition, Kohli et al. [86] found in a cross-sectional descriptive study of over 400 participants that 85% of newborns born to mothers with low serum ferritin levels (<50 µg/L) had low cord blood hemoglobin levels (<14 g/dL), suggesting a significant association between maternal and neonatal iron status. The findings in humans, however, are not always consistent. Shao et al. [87] found, across over 3000 pregnant women in China, that in the absence of severe maternal ID (serum ferritin < 13.6 µg/L), cord blood serum ferritin was not significantly affected, and the fetal iron needs could still be met under a milder maternal ID. These inconsistencies could be due to many confounding factors across individuals (i.e., for review [88]), one of which, we suggest, could be Pb.

The diet-induced decreases in the offspring hematological measures of ID were further corroborated by the significant decreases in the circulating blood Fe levels. Interestingly, the significant depletions in the blood Fe levels were not necessarily mirrored in the brain, confirming findings by others of region-specific depletions in Fe [89,90]. We observed in offspring that the cerebral cortex was more sensitive to iron depletion than the hippocampus, irrespective of the concentration of Pb exposure administered (0, 19, or 50 ppm Pb). When using an equivalent dietary paradigm in rats, Greminger et al. noted similar region-specific vulnerabilities in PND7 rat pups to iron depletion, as measured by reductions in cortical and hippocampal ferritin levels [55].

Studies have shown that iron depletion is associated with elevated BPb levels. Importantly, irrespective of whether IN or ID, an exposure to a higher Pb concentration in our model led to higher circulating and stored Pb levels than a lower Pb concentration. Specifically, in ID, a lower Pb exposure (19 Pb) increased the Pb burden in the blood, femur, and brain. In contrast, with a higher Pb exposure (50 Pb), however, the ID-associated increases in Pb burdens were only apparent in the Pb stores within the femur and in the brain but not within circulation. We suggest that in the context of a higher Pb exposure, this reflects a saturation of the Pb binding sites in red blood cells in whole blood. This saturation subsequently promotes a redirection of the remaining Pb into the plasma fraction of blood, which more readily exchanges metals with tissue [91,92,93,94]. Importantly, these findings further provide a potential mechanistic rationale for studies that found exacerbations in intellectual and motor deficits in gestationally Pb-exposed 6-month-old infants as a consequence of maternal ID (i.e., [40]).

The consequences of Pb, however, vary greatly across the population. It remains unclear to what extent impairments are driven by the net Pb concentration achieved within a given brain region, especially because the extent of region-specific Pb accumulation and its potential associated impairments can be variable depending on the context of the Pb exposure. Another challenge to predicting these impairments is that BPb levels are the generally accepted biomarker to estimate Pb risk [95]. In our study, we found that BPb levels are predictive of brain Pb burdens but only in the context of IDA. Under IN conditions, BPb levels are poorly correlated with the brain Pb burden. Interestingly, recent evidence suggests that the cumulative bone Pb burden was more strongly associated with cognitive function than circulating BPb levels [96]. These studies, however, were conducted in adults and none of them addressed the potential influence of iron status. Therefore, it would be interesting to see whether femoral Pb levels correlate with brain Pb levels, irrespective of gestational iron history.

Another interesting future direction would be to investigate the impact of iron supplementation on developmental Pb exposure. Rosado et al. found that iron supplementation was successful in improving iron status but not in diminishing BPb levels in exposed children [97]. However, Shah-Kulkarni et al. observed that the strength of the association between maternal BPb values late in pregnancy and cognitive development early in childhood was significantly affected by maternal dietary iron status [40]. Irrespective of the nature underlying these observations, the lack of a correlation between BPb and brain Pb in our iron-normal animals suggests that iron supplementation, as recommended by the World Health Organization [98,99], may or may not affect these correlations and thereby influence Pb risk assessments.

## 5. Conclusions

Overall, our study provides insight into the complex dynamics of divalent metal homeostasis during pregnancy and its impact on the developing offspring. Our animal exposure uniquely co-models ID and Pb exposure together to provide a “real world” context that affects thousands of pregnant women and their offspring. Specifically, our data showed that without knowledge of the underlying gestational history of iron levels or consideration of the iron status of the postnatal offspring, BPb levels are not informative and may not accurately facilitate an estimation of the brain Pb burden. The critical and novel implications of our study emphasize the need for a better understanding of the interactions between Fe and Pb during pregnancy and for improvements in the diagnostic indices used to assess children at risk.

## Figures and Tables

**Figure 1 nutrients-15-04101-f001:**
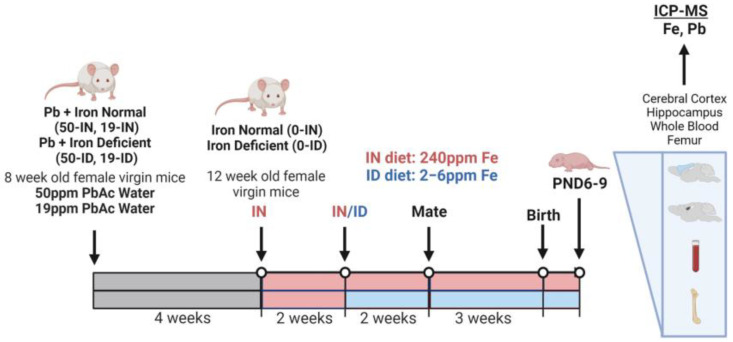
Schematic of animal exposure paradigm and micro-isolated tissues (Created with BioRender.com; accessed on 1 August 2023).

**Figure 2 nutrients-15-04101-f002:**
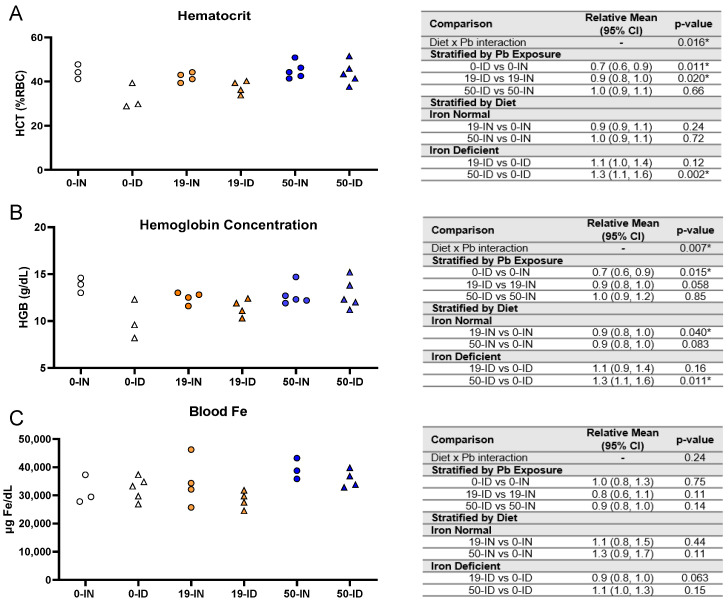
Effects of ID diet and Pb on maternal blood iron indices. Original data values are graphed for maternal hematocrits (**A**), hemoglobin concentrations (**B**), and venous blood Fe levels (**C**). Statistical analyses were performed using log-link gamma GLMs, separated by either iron deficiency or Pb exposure. The test for interaction was based on a single overarching log-link gamma GLM including both diet and Pb exposure plus their interaction; a significant interaction indicates evidence that the relative means for diet (ID vs. IN) differ by Pb exposure, and the relative means for Pb exposure (19 vs. 0, 50 vs. 0) differ by diet. Statistically significant differences are indicated with “*” in tables adjacent to graphs. Abbreviations: HCT = hematocrit; RBC = red blood cell; HGB = hemoglobin concentration; g/dL = grams per deciliter; µg/dL = micrograms per deciliter.

**Figure 3 nutrients-15-04101-f003:**
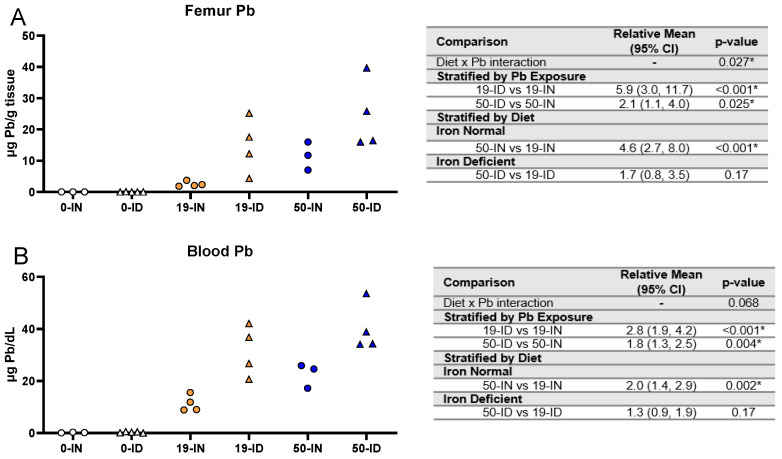
Effect of ID diet and Pb exposure on maternal blood and femoral Pb. Original data values are graphed for Pb levels in the femurs (**A**) and blood (**B**) of exposed dams. Statistical analyses were performed using log-link gamma GLMs, separated by either iron deficiency or Pb exposure and adjusted for sex and litter size. The test for interaction was based on a single overarching log-link gamma GLM including both diet and Pb exposure plus their interaction, as well as sex and litter size; a significant interaction indicates evidence that the relative means for diet (ID vs. IN) differ by Pb exposure, and the relative means for Pb exposure (50 vs. 19) differ by diet. Statistically significant differences are indicated with “*” in tables adjacent to graphs. Abbreviations: µg/g = micrograms per gram; µg/dL = micrograms per deciliter.

**Figure 4 nutrients-15-04101-f004:**
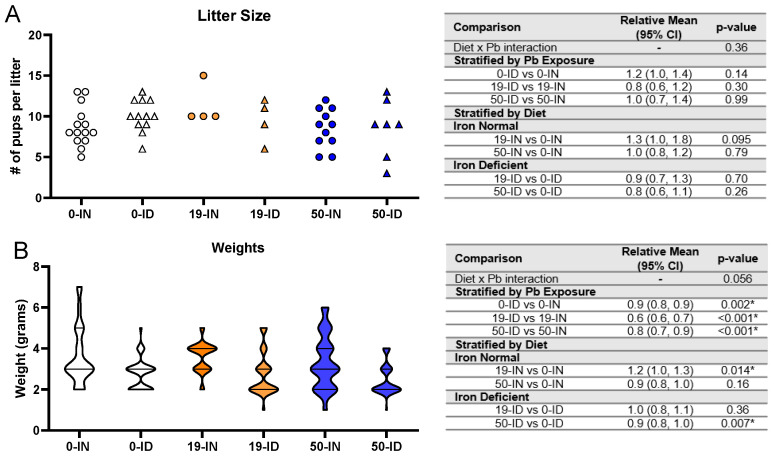
Effects of ID diet and Pb on litter size and offspring weights. Original data values are graphed for litter sizes (**A**) and weights of offspring (**B**). Statistical analyses were performed for both using log-link gamma GLMs, separated by either iron deficiency or Pb exposure, but weight analyses were additionally adjusted for sex and litter size. The test for interaction was based on a single overarching log-link gamma GLM including both diet and Pb exposure plus their interaction, as well as sex and litter size in weight analyses; a significant interaction indicates evidence that the relative means for diet (ID vs. IN) differ by Pb exposure, and the relative means for Pb exposure (19 vs. 0, 50 vs. 0) differ by diet. Within each treatment group is a minimum of 1 animal per sex across 4–14 (for litter size analyses) or 3–7 (for weights analyses) independent litters. Statistically significant differences are indicated with “*” in tables adjacent to graphs.

**Figure 5 nutrients-15-04101-f005:**
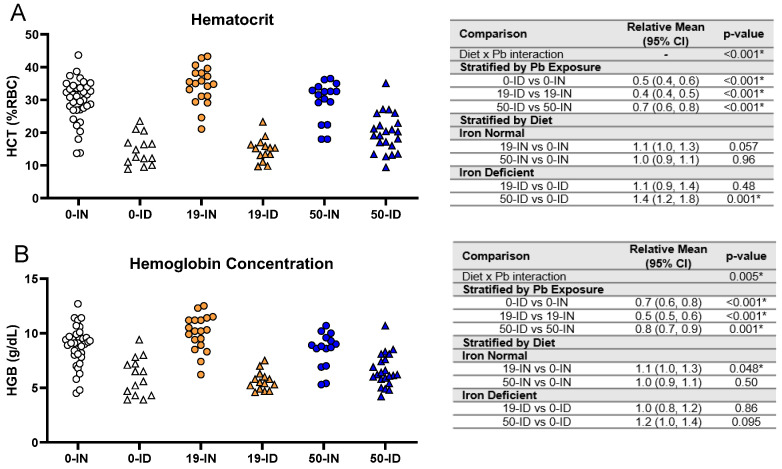
Effects of ID diet and Pb on offspring hematocrit and hemoglobin values. Original data values are graphed for hematocrits (**A**) and hemoglobin levels (**B**) in trunk blood of offspring. Statistical analyses were performed using log-link gamma GLMs, separated by either iron deficiency or Pb exposure, adjusted for sex and litter size. The test for interaction was based on a single overarching log-link gamma GLM including both diet and Pb exposure plus their interaction, as well as sex and litter size; a significant interaction indicates evidence that the relative means for diet (ID vs. IN) differ by Pb exposure, and the relative means for Pb exposure (19 vs. 0, 50 vs. 0) differ by diet. Within each treatment group is a minimum of 1 animal per sex across 3–7 independent litters. Statistically significant differences are indicated with “*” in tables adjacent to graphs. Abbreviations: HCT = hematocrit; RBC = red blood cell; HGB = hemoglobin concentration; g/dL = grams per deciliter.

**Figure 6 nutrients-15-04101-f006:**
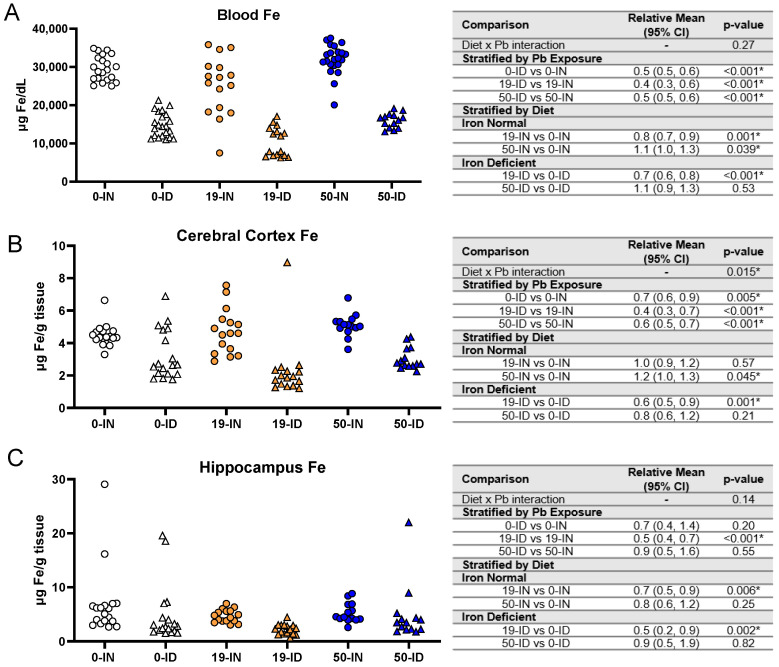
Effects of ID diet and Pb on offspring circulating and brain Fe levels. Original data values are graphed for circulating (**A**) and brain Fe levels in the cerebral cortex (**B**) and hippocampus (**C**) of offspring. Statistical analyses were performed using log-link gamma GLMs, separated by either iron deficiency or Pb exposure, adjusted for sex and litter size. The test for interaction was based on a single overarching log-link gamma GLM including both diet and Pb exposure plus their interaction, as well as sex and litter size; a significant interaction indicates evidence that the relative means for diet (ID vs. IN) differ by Pb exposure, and the relative means for Pb exposure (19 vs. 0, 50 vs. 0) differ by diet. Within each treatment group is a minimum of 1 animal per sex across 3–7 independent litters. Statistically significant differences are indicated with “*” in tables adjacent to graphs. Abbreviations: µg/dL = micrograms per deciliter; µg/g = micrograms per gram.

**Figure 7 nutrients-15-04101-f007:**
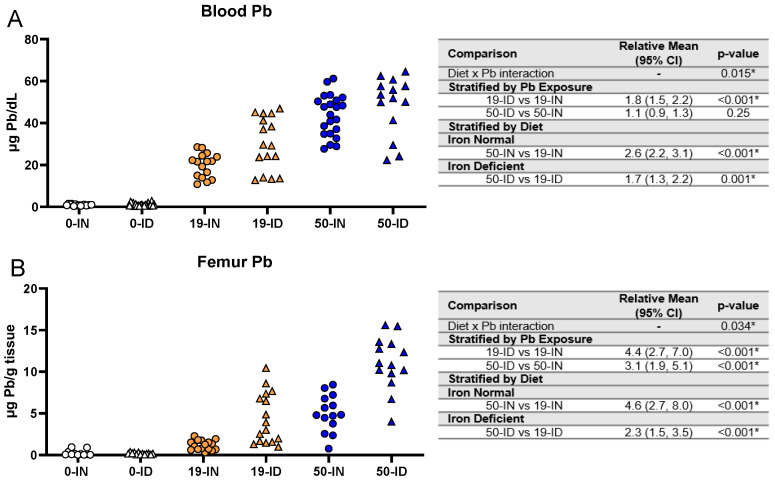
Effects of ID diet and Pb on offspring circulating and stored Pb levels. Original data values are graphed for circulating blood (**A**) and stored femoral (**B**) Pb levels in offspring. Statistical analyses were performed amongst the 19 ppm and 50 ppm Pb acetate-exposed groups using log-link gamma GLMs, separated by either iron deficiency or Pb exposure, adjusted for sex and litter size. The test for interaction was based on a single overarching log-link gamma GLM including both diet and Pb exposure plus their interaction, as well as sex and litter size; a significant interaction indicates evidence that the relative means for diet (ID vs. IN) differ by Pb exposure, and the relative means for Pb exposure (50 vs. 19) differ by diet. Within each treatment group is a minimum of 1 animal per sex across 3–7 independent litters. Statistically significant differences are indicated with “*” in tables adjacent to graphs. Abbreviations: µg/dL: micrograms per deciliter; µg/g = micrograms per gram.

**Figure 8 nutrients-15-04101-f008:**
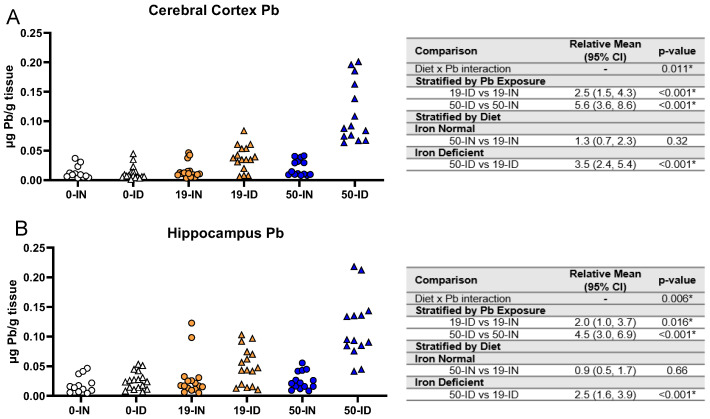
Effects of ID diet and Pb on offspring brain Pb burden. Original data values are graphed for brain Pb levels in the cerebral cortex (**A**) and hippocampus (**B**). Statistical analyses were performed amongst the 19 ppm and 50 ppm Pb acetate-exposed groups using log-link gamma GLMs, separated by either iron deficiency or Pb exposure, adjusted for sex and litter size. The test for interaction was based on a single overarching log-link gamma GLM including both diet and Pb exposure plus their interaction, as well as sex and litter size; a significant interaction indicates evidence that the relative means for diet (ID vs. IN) differ by Pb exposure, and the relative means for Pb exposure (50 vs. 19) differ by diet. Within each treatment group is a minimum of 1 animal per sex across 3–7 independent litters. Statistically significant differences are indicated with “*” in tables adjacent to graphs. Abbreviation: µg/g = micrograms per gram.

**Figure 9 nutrients-15-04101-f009:**
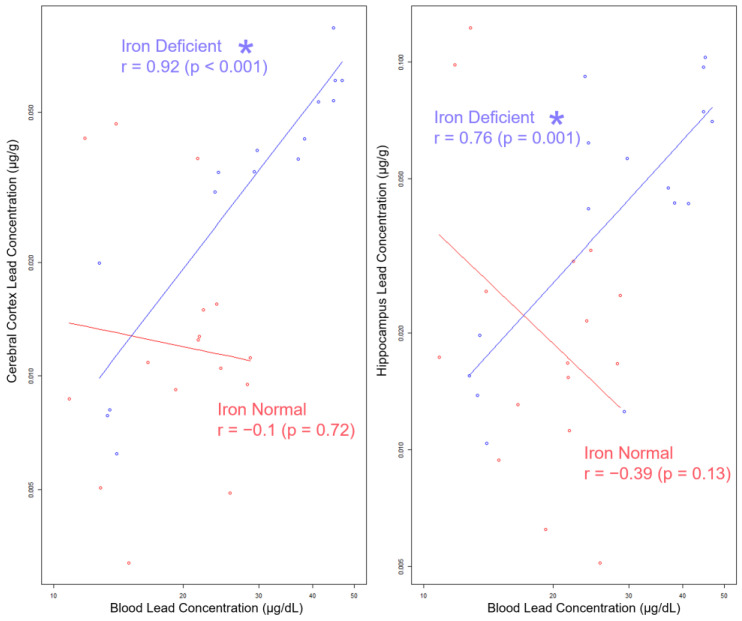
Log–log scale scatterplots of Pb concentrations in cerebral cortex and hippocampus by BPb concentration, stratified by diet. Least squares lines are superimposed on the data for each diet group, and the corresponding Pearson’s product–moment correlation coefficients (r) and *p*-values for the log-transformed Pb concentrations are annotated in the graph. Statistically significant differences are indicated with “*”. Abbreviations: µg/g = micrograms per gram; µg/dL = micrograms per deciliter.

## Data Availability

The data presented in this study are available on request from the corresponding author.

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
