# Peer review of "Maternal Iron Deficiency and Environmental Lead (Pb) Exposure Alter the Predictive Value of Blood Pb Levels on Brain Pb Burden in the Offspring in a Dietary Mouse Model: An Important Consideration for Cumulative Risk in Development"

_nutrients, 2023, doi:10.3390/nu15194101_

Round 1

Reviewer 1 Report

Thank you for the opportunity to read this well-written, clear and concise study. It was interesting and a pleasure to read.

The introduction is very complete and allows the reader to know the main topic of the research, informs about the purpose and importance of the work in the clinical field, and also answers the question posed in the scientific context. It includes previous works on the topic in question and makes clear the aspects to be detailed in the review, which constitutes the object of the proposed research. It explains the general problem of the research, includes previous work on the topic in question, and specifies the objective of the study.

The “methods” section is one of the most fundamental sections of a scientific article with these characteristics and must be reviewed by the authors:

- They must add a first subsection that specifically talks about the type of study.

- In the part where the sample is discussed, it is necessary to add a flow chart that explains the initial and total number of participants, exposing those who were excluded from the study.

- What were the inclusion and exclusion criteria?

- The Tissue Harvests, who collected them? The person who obtained said sample must be added in the subsection.

Lines 195-200: do not comply with the format of the entire manuscript. Please review and modify.

- Between lines 210-211 there is a space that should not appear.

- Lines 234-236: do not comply with the format of the entire manuscript. Please review and modify.

- Lines 267-368: do not comply with the format of the entire manuscript. Please review and modify.

- Lines 288-294: do not comply with the format of the entire manuscript. Please review and modify.

- Lines 334-339: do not comply with the format of the entire manuscript. Please review and modify.

- Lines 368-369: do not comply with the format of the entire manuscript. Please review and modify.

- Lines 383-389: do not comply with the format of the entire manuscript. Please review and modify.

The authors of this article have meticulously explained the results, providing relevant figures to what is explained in the text. On the other hand, they have provided a clear and complete discussion comparing their results with previous studies and arguing the existing differences. Furthermore, the limitations of this research are presented very clearly.

Finally, the references section should be reviewed, since it does not comply with the journal's standards.

Author Response

We like to thank you for your positive feedback and your detailed and thorough review of our manuscript.

We agree with your assessment of adding more methodological details and extended the description of the design and animal numbers in the material and methods (highlighted in yellow).  In addition, we also added a supplementary table outlining the assignment of dams and pups to specific treatment groups and measurements.

We clarified that we did not exclude any animals in our statistical analyses and named the individual who handled all animals and harvested all tissues.

We apologize for the inconsistent formatting and have now changed all the indentations appropriately.

We have also updated the reference library according to the journal's formatting requirements.

In summary, we hope that our additions address your initial concerns and questions and thank you again for your interest in our work.

Reviewer 2 Report

The presented paper is processed at a very good level and its focus is suitable for your journal. I have only a few minor comments. Keywords are repeated in the title. In the abstract I recommend briefly supplementing the data on the methodology of the experiment. Please give the number of the experiment permission decision. In the material and methodology section, please add the numbers of animals used, that is, the numbers of mothers used, the numbers of their young, the numbers of animals euthanized.

Author Response

We thank you for supporting our manuscript and have addressed your initial concerns in the following manner (all changes are highlighted in yellow):

We thank you for pointing out that the keywords were repeated in the title and we have now added unique keywords that are not stated in the title.

We extended the methodological information in the abstract in the confines of 200 words.

In line with your recommendation, which we very much appreciated, we have also extended the materials and methods section and added a detailed supplementary table that outlines the number of dams and pups designated for each treatment group and measurement. We also clarified that no animals were excluded for our statistical analyses.

In summary, we thank you again for your interest and hope that our comments and changes address your concerns.